

# Factors affecting genotyping success in giant panda fecal samples

Ying Zhu[1,2], Hong-Yi Liu[3], Hai-Qiong Yang[1,2], Yu-Dong Li[1,2] and He-Min Zhang[2,4]

[1] Sichuan Nature Resources Science Academy, Chengdu, Sichuan, China
[2] Sichuan Province Laboratory for Natural Resources Protection and Sustainable Utilization, Chengdu, Sichuan, China
[3] Co-Innovation Center for Sustainable Forestry in Southern China, College of Biology and the Environment, Nanjing Forestry University, Nanjing, Jiangsu, China
[4] China Conservation and Research Center for the Giant Panda, Dujiangyan, Sichuan, China

## ABSTRACT

Fecal samples play an important role in giant panda conservation studies. Optimal preservation conditions and choice of microsatellites for giant panda fecal samples have not been established. In this study, we evaluated the effect of four factors (namely, storage type (ethanol (EtOH), EtOH $-20$ °C, 2-step storage medium, DMSO/EDTA/Tris/salt buffer (DETs) and frozen at $-20$ °C), storage time (one, three and six months), fragment length, and repeat motif of microsatellite loci) on the success rate of microsatellite amplification, allelic dropout (ADO) and false allele (FA) rates from giant panda fecal samples. Amplification success and ADO rates differed between the storage types. Freezing was inferior to the other four storage methods based on the lowest average amplification success and the highest ADO rates ($P < 0.05$). The highest microsatellite amplification success was obtained from either EtOH or the 2-step storage medium at three storage time points. Storage time had a negative effect on the average amplification of microsatellites and samples stored in EtOH and the 2-step storage medium were more stable than the other three storage types. We only detected the effect of repeat motif on ADO and FA rates. The lower ADO and FA rates were obtained from tri- and tetra-nucleotide loci. We suggest that freezing should not be used for giant panda fecal preservation in microsatellite studies, and EtOH and the 2-step storage medium should be chosen on priority for long-term storage. We recommend candidate microsatellite loci with longer repeat motif to ensure greater genotyping success for giant panda fecal studies.

# INTRODUCTION

Sampling of feces has become a feasible and widely used method for researchers to obtain genetic data in the field, especially in genetic research on endangered animals. Such high usage is attributable to the sampling being convenient, random, non-invasive, and non-interfering to the animal's activity, and is facilitated by the abundance of samples (*Kohn & Wayne, 1997*). The research using fecal DNA includes documents on species identification (*Dalen, Gotherstrom & Angerbjorn, 2004*), sex determination (*Huber, Bruns & Arnold,*

Corresponding author
He-Min Zhang,
wolong_zhm@163.com,
wolong_zhm@126.com

2002), kinship and paternity (*Constable et al., 2001*), population genetic diversity (*Zhang et al., 2007*), adaptive variation (*Wan et al., 2006*), population genetic structure (*Zhu et al., 2013*), dispersal pattern (*Zhan et al., 2007*), population size (*Eggert, Eggert & Woodruff, 2003*), evolutionary history of species (*Chen et al., 2013*), mating system (*Garnier, Bruford & Goossens, 2001*), and disease information (*Zhang et al., 2012*).

Despite the many advantages as stated above, low quantity and poor quality of fecal DNA, often results in the failure of amplification and errors in microsatellite genotyping in such studies (*Taberlet et al., 1996*). This renders the conclusion of genetic studies unreliable and reduces the confidence in inferring such results for formulating management and conservation strategies (*Pompanon et al., 2005*). Several investigations suggest that careful choice of microsatellite loci and the method used for feces preservation could enhance the genotyping success and feasibility of the use of fecal samples in such studies (*Broquet, Ménard & Petit, 2007*; *Tende et al., 2014*).

A comparison of storage conditions has been made among oven-dried, frozen, and ethanol (EtOH) and buffer-preserved fecal samples from different species (reviewed in *Tende et al., 2014*). The conclusions have, however, been inconsistent, even for the same preservation medium, suggesting that the optimal storage medium varies with species, environmental conditions, and other factors (*Piggott & Taylor, 2003*). Moreover, it was reported that the optimal storage types were dependent on the storage period (*Murphy et al., 2002*; *Soto-Calderon et al., 2009*), which could be the reason for the varied degradation rate of fecal DNA in different preservation media. Such observations necessitate the assessment of the storage type for each new fecal study.

DNA fragment length and the microsatellite repeat motif present in it are known to greatly impact the amplification and microsatellite genotyping success (*Broquet, Ménard & Petit, 2007*). Most of the studies revealed that the amplification success decreased and the genotyping error rates increased with the fragment length and that the di-nucleotides were superior to longer repeat units (reviewed in *Broquet, Ménard & Petit, 2007*). In contrast, other studies have reported contradictory results, demonstrating that the allele length had no effect on the error rates (*Frantz et al., 2003*) or that the longer fragments were easier to be amplified than the shorter ones (*Whittier et al., 1999*). Furthermore, motifs with longer nucleotide repeats were associated with lower error rates compared to the di-nucleotide motifs (*Kruglyak et al., 1998*).

The giant panda (*Ailuropoda melanoleuca*) is an endangered species in China owing to habitat fragmentation. Fecal samples play an important role in its conservation studies, and they have been widely used in several genetics studies (*Chen et al., 2013*; *Wan et al., 2006*; *Zhan et al., 2007*; *Zhang et al., 2007*; *Zhu et al., 2013*). DNA obtained from fecal samples have been stored under dried conditions (*Wan et al., 2006*) and in EtOH (*Zhang et al., 2004*; *Zhang et al., 2012*). However, to our knowledge, evaluation of a long-term storage type for fecal DNA has not been conducted yet. Research on the effects of microsatellite fragment length and microsatellite motif has also been limited in giant panda fecal studies. The present study aimed to: (1) compare the performance of five storage conditions and three storage periods in the preservation of fecal DNA of giant panda; (2) determine the optimal method that produced the highest amplification success and lowest genotyping

errors of microsatellites; and (3) evaluate the effect of fragment length and repeat unit on the genotyping success at microsatellite loci. We believe that the information gained in the present study would be useful in guiding researchers to choose the most suitable preservation medium and microsatellite loci in giant panda fecal studies and allow for assessment of results from past studies.

## METHODS AND MATERIALS

### Sample collection and preservation

We obtained permission from the Conversation Base of China Research and Conservation Center for the Giant Panda (CRCCGP) and the China Giant Panda Protection and Management Office to collect all the samples and confirmed that we did not impact the animals during the sampling. Eighty-five fecal samples from 17 captive giant pandas housed in the Dujiangyan Giant Panda Conservation Base, were collected on March 20, 2015. The samples were collected from the outdoor house in the afternoon after giant pandas were recalled to their indoor house. Fresh samples (<12-hours-old) were collected using disposable gloves and transferred into Ziploc bags, with the sample information marked clearly on the bags. The samples were transported immediately to the laboratory after 2 h journey by car. The surface of each sample was peeled off with sterile tweezers, and the sample from each individual was mixed to avoid an uneven distribution of intestinal cells (*Kohn & Wayne, 1997*). The samples were then divided into five parts and transferred to five storage media, namely absolute EtOH, absolute EtOH at −20 °C, silica after 24-hours treatment with EtOH, which was designated as the "2-step storage medium" (*Tende et al., 2014*), DMSO/EDTA/Tris/salt buffer (DETs) buffer (*Frantzen et al., 1998*), or frozen at −20 °C. In addition, we also obtained blood samples from these 17 individuals for use as reference samples when comparing the microsatellite genotypes from the fecal samples. The blood samples were collected by a veterinarian during the routine medical examination of pandas for disease monitoring and were stored in a −80 °C freezer.

### DNA extraction and microsatellite amplification

Two-gram dried samples were weighed from each storage medium, transferred to individual 50-mL tubes containing 10 mL Tris/NaCl/EDTA/SDS (TNES) buffer and 80 μL 20 mg/mL proteinase K, and incubated overnight at 65 °C in a water bath. DNA from the fecal samples was extracted using the modified phenol-chloroform extraction method described by *Wan et al. (2006)*. To evaluate the effect of storage time for each storage condition, DNA was extracted after one, three and six months, respectively. DNA from blood was extracted with the Invitrogen blood extraction kit (Invitrogen, Carlsbad, CA, USA) following the protocol prescribed by the kit manufacturer. A total of 272 DNA samples (extracted from 255 fecal and 17 blood samples) were used in the present study.

Nine microsatellite loci (Gp4, Aime10, Aime16, Panda22, Panda25, Panda29, Gpz6, Gpl29, and Gpl60 (*Huang et al., 2015*; *Shen et al., 2007*; *Wu et al., 2009*; *Zhang et al., 2009*)) with good performance in amplification and genotyping were chosen for evaluation in the present study. All the forward primers were mixed with a fluorescent M13 primer (5′-CACGACGTTGTAAAACGAC-3′; Li-COR Inc., Lincoln, NE, USA). The amplifications

were carried out in a 10-μL reaction mixture that contained 10X buffer with 25 mM MgCl$_2$, 0.2 mol/L of each primer, 1 μL of 1 μM Infrared Dye Phosphoramidite-labeled M13 primer, 0.2 mmol/L dNTPs, 0.25 U r-Taq (TaKaRa Ltd, Dalian, China), and approximately 10 ng of the DNA template. For the fecal DNA samples, 0.1 μL BSA was added to each reaction mixture. All the amplifications were performed using a touch-down profile which began at 95 °C for 5 min, followed by 15 touchdown cycles of 95 °C for 30 s, 62.5−52 °C for 30 s, and 72 °C for 30 s, decreasing by 0.7 °C to the annealing temperature and 25 cycles with an annealing temperature of 52 °C. A final amplification step was carried out at 72 °C for 5 min. Each sample at each locus was amplified three times and an allele was accepted when it was detected at least twice.

DNA extraction and PCR amplification were conducted in separate rooms and positive and negative controls were included in all the experiments.

## Data analysis
### Comparison of storage type and storage time
The mean microsatellite amplification success was the proportion of all the microsatellite loci that were successfully amplified for each storage type and each storage time. We further analyzed the genotyping errors for microsatellites; these were the allelic dropouts (ADO, one allele of a heterozygote was lost during the amplification) and the false alleles (FA, the allele generated during PCR due to a slippage artifact). We determined the occurrence of ADO and FA by comparing the genotype produced by the fecal samples with the genotype obtained from the blood samples. ADO and FA rates were calculated according to the equations (2) and (4) described in *Broquet & Petit (2004)*.

We analyzed three dependent variables: amplification success, ADO, and FA rates using a repeated measure general linear model, with preservation methods as the "between-subject variable" and storage time as the "repeated element" followed by a Tukey's multiple comparison test. If the effects between the factors were significant, we interpreted simple main effects using least significant differences analysis to separate the means.

### Comparison of fragment length and repeat motif at microsatellite loci
We classified nine microsatellite loci by their fragment lengths and obtained 3 grades: Grade I included Gp4 and Panda25, which had a product size smaller than 150 bp; Grade II contained Panda29 and Gpl29, with product sizes between 150 and 200 bp; Grade III had Aime10, Aime16, Panda22, Gpz6, and Gpl60 with product sizes between 200 and 300 bp. Moreover, the loci were classified into three groups with di-(Gp4, Aime10, and Aime16), tri-(Panda22, Panda25, and Panda29), and tetra-nucleotide repeats (Gpz6, Gpl29, and Gpl60), respectively. We compared the effects of fragment length and repeat motif on the amplification success, and the ADO and FA rates using a multivariate general linear model with the amplification success, ADO, and FA rates as dependent variables, fragment length and repeat motif as fixed factors, and storage type as covariate. We adopted the data obtained at the storage time of one month to avoid the time effect. All the statistical tests were performed using SPSS 20.0 (IBM, Chicago, IL, USA).

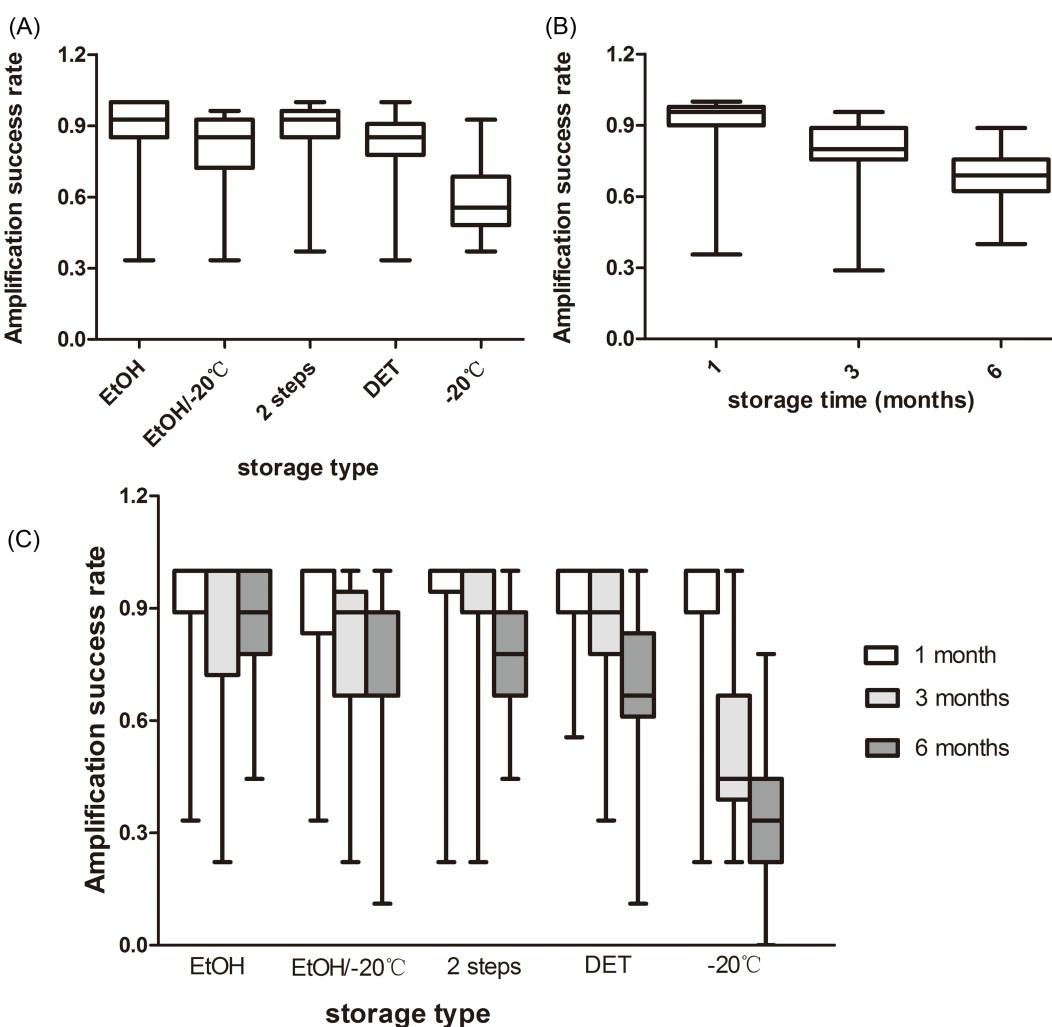

**Figure 1  Rate of amplification success at giant panda microsatellite loci amplified from fecal DNA stored in five storage types at three storage intervals.** (A) The average amplification success across the three storage times; (B) The average amplification success across the five storage types; (C) The average amplification success for five storage types at three storage intervals. The whiskers show the values of the minimum, 2.5th percentile, median, and 97.5th percentile.

## RESULTS

### Evaluation of storage type and storage time on the amplification success at microsatellite

The storage type had an effect on the microsatellite amplification success ($F_{4,80} = 9.976$, $P < 0.001$). The samples stored in EtOH showed the best performance (89.1%), followed by those stored in the 2-step storage medium (87.6%), DET/EtOH/$-20\,°C$ (80.4%), and at $-20\,°C$ (57.8%; Fig. 1A). The amplification success in the freezing type was significantly lower than in the other four storage types and there were no significant differences between any other comparisons (Table S1).

We observed that storage time had an effect on the amplification success, as well ($F_{2,80} = 61.306$, $P < 0.001$). As the storage time increased, the amplification success decreased

from 90.7% to 68.0% or by approximately 23% (Fig. 1B). Amplification successes obtained from three storage times were different from each other (Table S2).

There was an interaction between the effects of the storage time and storage types on the microsatellite amplification success ($F_{8,80} = 9.099$, $P < 0.001$), revealing that the optimal storage type varied among the different storage times (Fig. 1C). The highest amplification success in the first month was obtained from EtOH and the 2-step storage medium (92.8%). The 2-step storage medium and EtOH showed the best performance in the samples stored for three months (90.8%) and six months (87.6%), respectively. The amplification success in the samples stored under freezing conditions was least at all the three storage times (Fig. 1C).

Simple main effects analysis showed that there was statistically significant difference in the amplification success among the five storage types when the samples were stored for three months ($F_{4,80} = 9.590$, $P < 0.001$) and six months ($F_{4,80} = 18.970$, $P < 0.001$), but not in those stored for one month ($F_{4,80} = 0.240$, $P = 0.915$). Specifically, at the end of the three months, the amplification success of the freezing type was significantly lower than the other four storage types and the other four types did not show any statistical difference (Table S1). As the storage time increased to six months, the amplification success in the freezing type was still significantly lower than in the other four storage types, and EtOH storage had a significantly higher amplification success than the EtOH/$-20\,°C$, DET, and $-20\,°C$ storage, but not the 2-step storage (Table S1). There were no significant differences between any other comparisons (Table S1).

We also compared the amplification success among the three storage times for each storage type (Table S2). There was significant difference in the amplification success between the storage times when the samples were stored in EtOH/$-20\,°C$ ($F_{2,32} = 9.276$, $P = 0.001$), DET ($F_{2,32} = 23.543$, $P < 0.001$) and $-20\,°C$ ($F_{2,32} = 37.535$, $P < 0.001$), but not in EtOH ($F_{2,32} = 1.233$, $P = 0.305$) and 2-step storage ($F_{2,32} = 2.408$, $P = 0.092$). This signified the variation in the decline of the amplification success among the different storage types. The amplification success in the samples stored at $-20\,°C$ decreased by 60% from 1 month to 6 months; for those stored in DET, EtOH/$-20\,°C$, 2-step storage medium, and EtOH, it decreased by 27.6, 20, 15, and 6%, respectively.

## Evaluation of storage type and storage time on genotyping errors at microsatellites

None of the three terms, the storage type, storage time, and the interaction between them, had an effect on the FA rates (storage time: $F_{2,80} = 1.365$, $P = 0.261$; storage type: $F_{4,80} = 0.360$, $P = 0.836$; storage time × storage type: $F_{8,80} = 0.716$, $P = 0.677$). With respect to the ADO rate, only the storage type was significant (storage time: $F_{2,80} = 3.066$, $P = 0.052$; storage type: $F_{4,80} = 6.435$, $P < 0.001$; storage time × storage type: $F_{8,80} = 1.127$ $P = 0.354$). Specifically for the storage type, the samples stored in EtOH showed the lowest ADO rate, followed by EtOH/$-20\,°C$, 2-step storage medium, DET, and $-20\,°C$ (Table 1). The ADO rate in the freezing type was significantly higher than in the other four storage types and the other four storage types showed no significant differences among them (Table S3).

**Table 1 Allele dropout and false allele rates over nine microsatellite loci among the five storage types in samples preserved for one, three and six months.**

| | | EtOH | EtOH/−20°C | 2-step storage medium | DET | −20°C | Across types |
|---|---|---|---|---|---|---|---|
| ADO | 1 month | 1% | 3% | 7% | 4% | 8% | 5% |
| | 3 months | 1% | 5% | 0 | 7% | 30% | 9% |
| | 6 months | 8% | 5% | 11% | 11% | 30% | 13% |
| Across time[a] | | 3% | 4% | 6% | 7% | 23% | 8.7% |
| FA | 1 month | 7% | 7% | 10% | 8% | 6% | 8% |
| | 3 months | 7% | 10% | 8% | 9% | 14% | 9% |
| | 6 months | 5% | 6% | 5% | 4% | 12% | 6% |
| Across time | | 6% | 7% | 7% | 7% | 11% | 7.8% |

**Notes.**

ADO and FA are the abbreviations for allele dropout and false allele, respectively.

[a]Denotes that the storage type had an effect on the average amplification success over time.

### Evaluation of fragment length and repeat motif on amplification success and genotyping errors at the microsatellites

Fragment length had no effect on microsatellite amplification success, ADO and FA rates (amplification success: $F_{2,37} = 2.057$, $P = 0.142$; ADO: $F_{2,37} = 0.349$, $P = 0.707$; FA: $F_{2,37} = 2.343$, $P = 0.110$, Table S4).

The effect of repeat motif on the amplification success was not significant whereas its effects on the ADO and FA rates were significant (amplification success: $F_{2,37} = 0.326$, $P = 0.724$; ADO: $F_{2,37} = 8.468$, $P = 0.001$; FA: $F_{2,37} = 45.347$, $P < 0.001$, Fig. 2). Tri-and tetra-nucleotide loci showed lower ADO and FA rates than di-nucleotide loci ($P < 0.01$ in both cases, Fig. 2, Table S5) whereas the FA rate and ADO rate from tri-and tetra-nucleotide loci were similar ($P > 0.05$ in both the cases, Fig. 2, Table S5).

There was no interaction between the effects of the fragment length and repeat motif on the microsatellite amplification success, ADO, and FA rates (amplification success: $F_{2,37} = 2.742$, $P = 0.078$; ADO: $F_{2,37} = 0.945$, $P = 0.398$; FA: $F_{2,37} = 2.166$, $P = 0.129$).

## DISCUSSION

### Evaluation of the storage type and storage time

In the present study, we found that both the sample storage type and storage time influenced the microsatellite amplification and the storage type affected the genotyping reliability.

Methods for attempting to prevent degradation of fecal DNA samples include storing them in the preservation medium containing high salt concentration buffer, quick desiccation, or keeping the samples under low temperature conditions (reviewed in *Tende et al., 2014*). The present study showed that low temperature (−20 °C) was the least effective method for preserving the giant panda fecal DNA compared to the quick desiccation (EtOH and 2-step storage), high salt concentration (DET), and a combination of quick desiccation and low temperature (EtOH/−20 °C) methods, owing to lowest microsatellite amplification success and highest ADO rate ($P < 0.05$). Freezing was also found to be the least appropriate method in studies of Eurasian badger (*Frantz et al., 2003*), black and sun bear (*Wasser et*

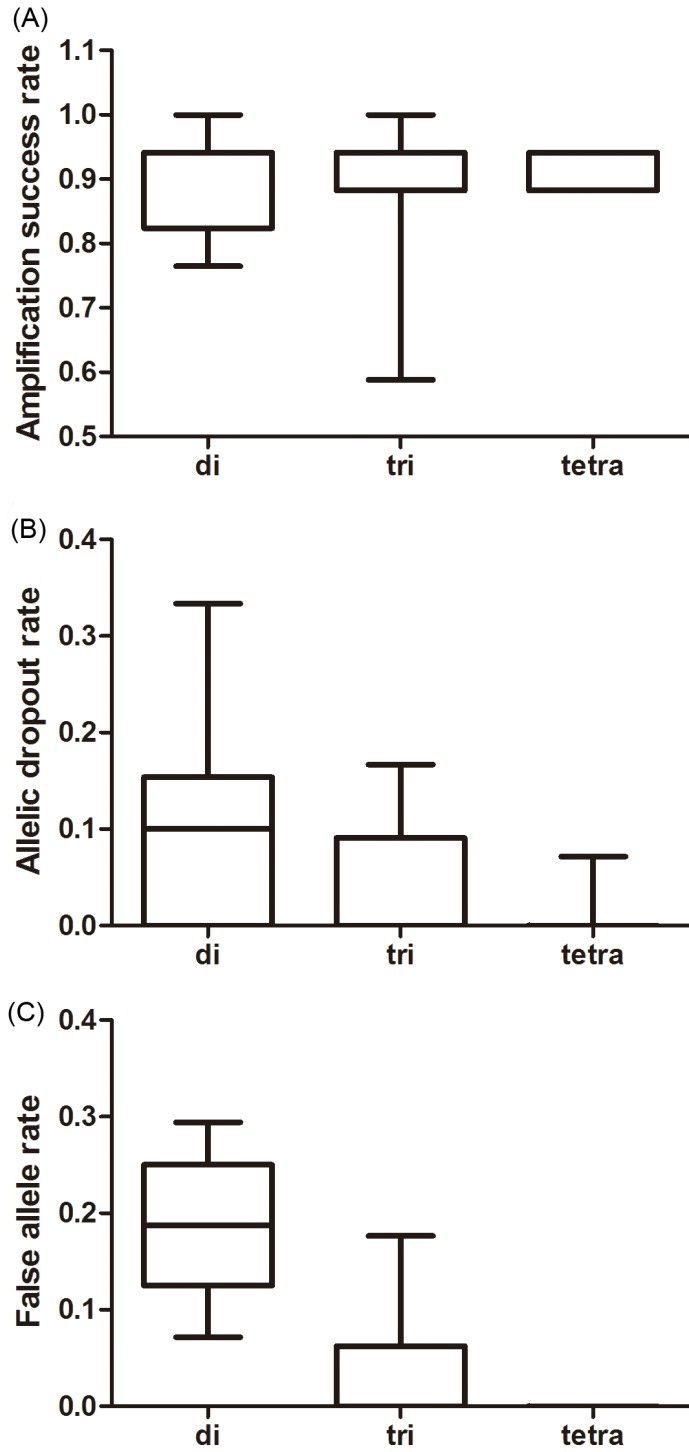

**Figure 2** **Average amplification success rate (A), allelic dropout (B), and false allele rates (C) for di-, tri-, tetra-nucleotide loci obtained from giant panda fecal DNA.** The whiskers show the values of the minimum, 2.5th percentile, median, and 97.5th percentile.

*al., 1997*), wolf (*Santini et al., 2007*), and baboons (*Frantzen et al., 1998*). We did not dry the samples prior to −20 °C preservation; therefore, the high moisture in the samples might have speeded the degradation in our study.

No significant differences were observed among the other four storage types, i.e., EtOH, EtOH/−20 °C, 2-step storage medium, and DET, based on the average amplification success and FA and ADO rates ($P > 0.05$). The samples stored in EtOH showed the highest microsatellite amplification success but the lowest ADO and FA rates. Similar to our results, EtOH scored the highest in the study on brown bear (*Murphy et al., 2002*), Eurasian badger (*Frantz et al., 2003*), lion (*Tende et al., 2014*), or was equally high as the other methods in the study on primate scats (*Whittier et al., 1999*), gorilla (*Roeder et al., 2004*), and coyote (*Panasci et al., 2011*), although the preservation methods evaluated in these studies were different. These findings revealed that EtOH was a good preservation buffer for fecal samples, irrespective of the diet of the species and this method could be used in either tropical or temperate weather (*Roeder et al., 2004*; *Tende et al., 2014*).

As observed in the present study, the 2-step storage also worked well in the preservation of giant panda fecal samples based on slightly lower amplification success (1%) and slightly higher ADO rate (3%) relative to the EtOH-preserved samples (Fig. 1; Table 1; Table S1; Table S3). Similar to our results, *Tende et al. (2014)* reported that the lion feces stored in the 2-step storage medium had lower amplification success rate than that preserved in EtOH. The 2-step storage medium was a combination of EtOH and silica, which was soaked in EtOH for a short period prior to desiccation by silica. We could not comment on the effectiveness of silica alone in storing the giant panda fecal samples compared to the 2-step storage medium owing to the lack of data on silica preserved samples in our study. However, in the studies on gorilla and chimpanzee feces (*Nsubuga et al., 2004*; *Roeder et al., 2004*) and the tiger feces (*Reddy et al., 2012*), the 2-step storage medium was superior to silica alone for producing more DNA, but outperformed or acted equally well as EtOH. The high-quality amplification obtained from the samples preserved in the 2-step storage medium might be attributed to two types of desiccation. We presumed that the 2-step storage medium would perform better than the silica alone in preserving the giant panda fecal DNA since EtOH was more effective than this medium in desiccating the samples in our case.

The other two storage types, EtOH at −20 °C and DET, would not be recommended for microsatellite fecal studies in giant panda, because of the great decrease in the amplification success during the six-month storage. Although it was revealed that EtOH at −20 °C (*Santini et al., 2007*) and DET (*Frantz et al., 2003*; *Frantzen et al., 1998*; *Panasci et al., 2011*) worked best in previous studies, these studies did not evaluate the variation of long term storage for these methods. Thus, we believe that these two methods might be useful for short-term fecal sample preservation. Fecal samples preserved in EtOH at −20 °C performed worse than those preserved in EtOH with the storage time increased. We speculated that the variation in the temperature (repeated freezing and thawing) might have led to the DNA degradation, which was supported by other studies (*Ross, Haites & Kelly, 1990*; *Shao, Khin & Kopp, 2012*).

As in other studies (*Murphy et al., 2002*; *Santini et al., 2007*; *Soto-Calderon et al., 2009*), we found that the storage time had a negative effect on the microsatellites amplification

success, suggesting that the preservation medium could only retard but not stop the degradation of fecal DNA by endonucleases. We recommended that fecal DNA should be extracted as soon as the samples are collected in order to obtain high DNA quality, to increase the accuracy of microsatellite genotyping, and to decrease the associated cost. Furthermore, our results indicated that the greatest decrease in microsatellite amplification success began at three months, but as we lacked data on genotyping from different months, we could not evaluate the relationship between the degradation rate and the storage time. Overall, it was indicated from our study that the samples stored in EtOH and the 2-step storage medium were more stable than those stored in EtOH/−20 °C, DET, and at −20 °C, since they had a slight change in microsatellite amplification success among the three storage times whereas the amplification success in the other three storage types decreased greatly with time. Thus, for microsatellite studies using samples stored for longer periods, EtOH and the 2-step storage medium might be a better choice.

### Evaluation of fragment length and repeat motif for microsatellites

For microsatellite studies, apart from selecting an appropriate storage type, careful choice of microsatellite loci could optimize the amplification and genotyping success of the fecal samples (reviewed in *Broquet, Ménard & Petit, 2007*). *Broquet, Ménard & Petit (2007)* found that amplification success decreased with the fragment length for microsatellites whereas the genotyping error rate increased. In the present study, we did not detect a linear relationship between the fragment length and amplification success or genotyping error rates, which might be due to the limited number of loci analysed by us.

Furthermore, it was reported that shorter repeat units produced lower ADO rates relative to the longer ones (*Broquet, Ménard & Petit, 2007*), which was opposite to our results. Our findings revealed that the highest FA rates were also obtained from the di-nucleotide microsatellite loci. *Broquet, Ménard & Petit (2007)* did not summarize the relationship between the repeat motif and FA rate owing to lack of data. False alleles were mostly caused in the process of auto-calling; therefore, it was normal that FA rate of di-nucleotide loci was higher because they were prone to slippage during PCR, which caused misscoring (*Pompanon et al., 2005*). Based on our findings, we recommended that it would be better to choose microsatellites with longer repeat motifs in fecal studies to obtain higher genotyping success.

## CONCLUSION

The findings of the present study revealed that fecal storage method should be carefully chosen in microsatellite studies relating to individual identification, kinship and paternity, and population size evaluation due to the effects of several factors. In fact, for short experiments, all the storage types except freezing could be useful for giant panda fecal sample preservation, owing to their relatively high amplification success (>80% for all the cases) and low genotyping error (<8% for all the cases). In contrast, for long-term storage, EtOH and the 2-step storage medium should be chosen for preservation of the fecal samples. However, we should also consider the transportation, expense, and other practical factors. Compared to EtOH, the 2-step storage medium has no problem with leaking and limitation of

air transportation. Nevertheless, it might cause contamination during the process of transferring the samples from EtOH to silica. We recommend that fecal DNA should be extracted as soon as possible because of the increase in degradation with time. The genotyping success of microsatellite loci could be enhanced by careful choice of the loci with longer repeat unit.

## ACKNOWLEDGEMENTS

We thank Wei M for the suggestion of fecal samples collection, Deng XX for blood collection, Liu Y, Dong JH, Xiong LB, Xu X, Liu HT, Wang YF, Tang C, Zheng YM, and Liu HT for the fecal samples collection, and Dr. He K for reviewing the manuscript.

### Funding

The present work was supported by a grant from the Science and Technology Department of Sichuan Province (2017TJPT0031) and a grant from the Basic Scientific Research Program of Sichuan Province for Public Welfare Institute. The funders had no role in study design, data collection and analysis, decision to publish, or preparation of the manuscript.

### Grant Disclosures

The following grant information was disclosed by the authors:
Science and Technology Department of Sichuan Province: 2017TJPT0031.
Basic Scientific Research Program of Sichuan Province for Public Welfare Institute.

### Competing Interests

The authors declare there are no competing interests.

### Author Contributions

- Ying Zhu performed the experiments, analyzed the data, contributed reagents/materials/analysis tools, wrote the paper, prepared figures and/or tables, reviewed drafts of the paper.
- Hong-Yi Liu analyzed the data, reviewed drafts of the paper.
- Hai-Qiong Yang performed the experiments, contributed reagents/materials/analysis tools.
- Yu-Dong Li performed the experiments.
- He-Min Zhang conceived and designed the experiments, reviewed drafts of the paper.

### Animal Ethics

The following information was supplied relating to ethical approvals (i.e., approving body and any reference numbers):

All the samples used in our study were collected from captive pandas which were housed in the Dujiangyan Giant Panda Conversation Base of China Research and Conservation Center for the Giant Panda (CRCCGP). Blood samples were obtained with permission from

the China Giant Panda Protection and Management Office during the routine medical examinations. We obtained permission from the CRCCGP to collect fecal samples and confirmed that we did not impact the animal during the sampling.

## Supplemental Information

Supplemental information for this article can be found online at http://dx.doi.org/10.7717/peerj.3358#supplemental-information.

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
