# Peer review of "Factors affecting genotyping success in giant panda fecal samples"

_PeerJ, doi:10.7717/peerj.3358_

## Round 0.1 · original submission · Minor Revisions

· Academic Editor

Minor Revisions

Many thanks for your submission. Having received two reviews, reviewer 2 has some major concerns about the methods used. Unless the experiment was to be completely repeated it is difficult to rectify at this stage. However they should be acknowledge in the manuscript or rebutted in the letter to the editor. I do not believe the corrections to the manuscript will take long and therefore have suggested minor corrections.

·

Basic reporting

The paper is well structured and well-written throughout. Aside from a number of minor language errors and improvements to wording (listed below), the language is clear and unambiguous and the figures well-presented.


Corrections to English:
Line 22 - replace ‘…lowest microsatellites averaged…’ with ‘…lowest average…’

Line 60 - replace ‘…repeat motif in it…’ with ‘…microsatellite repeat motif…’

Line 79 - remove comma (,)

Line 86 - animals

Line 87 - Conservation

Line 89 - ‘hours’ (and throughout paper)

Line 97 - ‘…reference samples…’

Line 113 - ‘…Li-COR Inc., ‘

Line 116 - ‘…TaKaRa Ltd, ‘

Line 136 - ‘…a Tukey’s multiple…’

Line 213 - ‘Methods for attempting to prevent degradation of fecal DNA samples include storing them…’

Line 215 - delete ‘only’

Line 219 - replace ‘most inappropriate method in the study of…’ with ‘least appropriate method in studies of…’

Lines 247-252 - Any idea why storing at -20 in EtOH would be worse than EtOH unfrozen??

Line 272 - replace ‘detected’ with ‘analysed’

Line 277 - replace ‘artificial reading’ with ‘auto-calling’

Line 280 - ‘…choose microsatellites with longer repeat motifs in fecal studies…

Line 282 - ‘The findings of the present study revealed that fecal storage method should be carefully chosen in microsatellite studies relating to…’

Figure 1 - I’m concerned that the annotation on the two upper graphs will be too small to read.

Figure 1 legend: - ‘Rate of amplification success at giant panda microsatellite loci…at three storage intervals.’

Experimental design

The research questions have been clearly stated and the subsequent experimental design is appropriate to the stated objectives of the study. The method includes an appropriate number of controls and is described in sufficient detail (except one point listed below) to enable replication.

Line 102 - How were samples dried? (temperature, duration, heat source etc.)

Validity of the findings

The conclusions have been clearly stated and are supported by the data. The only exception to this is as follows:

Line 163 - The authors indicate that there was a difference between the performance of EtOH and the 2-step methods at months 3 and 6. However below (lines 176-177) it states that there is no difference. This discrepancy should be clarified.

Additional comments

This was a well presented, well designed piece of research that comprehensively addresses a valid research question. The findings will no doubt help improve the quality of data resulting from non-invasive fecal surveys of giant panda and other species. If the authors are able to address the points listed above I am happy to recommend the paper for publication.

Reviewer 2 ·

Basic reporting

The article is coherent, well constructed, and the English is very good. I have a major methodological concern detailed below.

A few minor matters of basic reporting:

The opening paragraph should be trimmed. The number of hits in Web of Science is not relevant.

Please change Figure 1 to boxplots instead of bars. Boxplots will present the variation in your data much more clearly.

Have the raw microsatellite genotypes been made available?

Experimental design

Am I correct that you dried the frozen samples prior to extraction, but did NOT dry the samples stored at room temperature (line 221)? If this is the case, I am very much concerned about the validity of your result that freezing and EtOH/-20C is less effective than other preservation methods. The effect of drying on only some samples could have a profound affect on your results. Please explain. Drying is not necessary.

Also, please explain your decision to freeze at -20C instead of -80C.

How long did you store the fecal samples in silica during 2-step preservation? Did you change the silica if/when it changed color (i.e. the silica stopped absorbing moisture)? Not doing will reduce its effectiveness.

NEVER use ziploc bags to store fecal samples. They are not sterile, they are not reliably airtight, and they tear easily. Also, unless gloves are individually packaged and sterile, fecal samples should NOT be handled. Use a disposable sterile implement to avoid human contamination. I would have major concerns about contamination for any population genetic study. Use sterile 50ml or 15 ul tubes. Because you are only looking at amplification ADO and FA rates, it probably has not affected your results, but this is not good practice.

Validity of the findings

I find it peculiar that freezing after storage in EtOH would have a negative effect on amplification, especially over time. The beneficial effect of freezing seems well established both in the literature and the conservation biology community (it is also something I have observed with primate fecal samples). In your discussion, please explain why DNA would be subject to less degradation at higher temperatures. As I mention above, I am concerned about the drying effect.

Do you expect the diet of captive pandas to be the same as that of free-ranging pandas? Diet can have a strong effect on amplification success.

3 amplifications are enough to determine if an allele is heterozygous, but not homozygous from feces. You need at least 5 replicates for the latter (Hansen et al. 2007. Effects of genotyping protocols on success and errors in identifying individual river otters (Lontra canadensis) from their faeces. Molecular Ecology Notes.)

Are the differences reported on lines 233-235 (1% and 3%) significant? Fig 1 and Table 1 do not make this clear.

---

## Round 0.2 · accepted · Accept

· Academic Editor

Accept

Many thanks for making the corrections and providing a clear rebuttal letter. I look forward to seeing this published online.